# Application of the Holomorphic Tauc-Lorentz-Urbach Function to Extract the Optical Constants of Amorphous Semiconductor Thin Films

Manuel Ballester [1], Marcos García [2], Almudena P. Márquez [3], Eduardo Blanco [2], Susana M. Fernández [4], Dorian Minkov [5], Aggelos K. Katsaggelos [6], Oliver Cossairt [1,6], Florian Willomitzer [7] and Emilio Márquez [2,*]

1 Department of Computer Sciences, Northwestern University, 633 Clark St, Evanston, IL 60208, USA
2 Department of Condensed-Matter Physics, Faculty of Science, University of Cadiz, 11510 Puerto Real, Spain
3 Department of Mathematics, College of Engineering, University of Cadiz, 11510 Puerto Real, Spain
4 Photovoltaic Solar Energy Unit, Energy Department, CIEMAT, Avenida Complutense 40, 28040 Madrid, Spain
5 College of Energy and Electronics, Technical University of Sofia, 2140 Botevgrad, Bulgaria
6 Electrical and Computer Engineering Department, Northwestern University, Evanston, IL 60208, USA
7 Wyant College of Optical Sciences, University of Arizona, Tucson, AZ 85721, USA
* Correspondence: emilio.marquez@uca.es

**Abstract:** The Tauc–Lorentz–Urbach (TLU) dispersion model allows us to build a dielectric function from only a few parameters. However, this dielectric function is non-analytic and presents some mathematical drawbacks. As a consequence of this issue, the model becomes inaccurate. In the present work, we will adopt a procedure to conveniently transform the TLU model into a self-consistent dispersion model. The transformation involves the integration of the original TLU imaginary dielectric function $\epsilon_2$ by using a Lorentzian-type function of semi-width, $\Gamma$. This novel model is analytic and obeys the other necessary mathematical requirements of the optical constants of solid-state materials. The main difference with the non-analytic TLU model occurs at values of the photon energy near or lower than that of the bandgap energy (within the Urbach absorption region). In particular, this new model allows us to reliably extend the optical characterization of amorphous-semiconductor thin films within the limit to zero photon energy. To the best of our knowledge, this is the first time that the analytic TLU model has been successfully used to accurately determine the optical constants of unhydrogenated *a*-Si films using only their normal-incidence transmission spectra.

**Keywords:** amorphous semiconductors; dielectric function; optical properties; Tauc–Lorentz model; Tauc–Lorentz–Urbach model; thin-film characterization

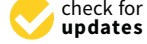



## 1. Introduction

The fundamental optical constants of solid-state materials are two mathematical functions that quantify the interaction of matter with the electromagnetic field of light waves. Despite their misleading name, the so-called optical 'constants' vary over the electromagnetic spectrum, and hence, they must be found at each vacuum wavelength/photon energy. It is essential to accurately calculate the dispersive optical constants of a solid-state material in order to consider it for diverse applications. These include the device design and characterization and the fabrication-process control of optical thin-film coatings. Furthermore, they play an essential role in the vast field of the semiconductor-device industry.

On the other hand, to gain insight into the the optical properties of thin layers and bulk materials knowledge regarding a pair of dimensionless frequency-dependent optical constants is required. These can be either the refractive index, *n*, and extinction coefficient, *k*, or the real and imaginary parts of the complex dielectric constant, $\epsilon_1$ and $\epsilon_2$, respectively. The optical properties of certain groups of solid materials in nature possess similar characteristic features, and permit simple optical models to give an account of their two dispersive optical constants over a particular spectral range. These dispersion models

depend only upon very few parameters to be evaluated for each specific material. Optical models facilitate the optical-constant determination since it only requires a limited number of free parameters. This idea is of utmost importance for the aim of the present work, as will be shown.

The optical parameters must meet some requirements if we desire to reproduce the dispersive optical constants of solid-state materials. Combining these optical constants, we can form a full complex function, namely $n + ik$ or $\epsilon_1 + i\epsilon_2$. This complex function is susceptible to being imposed upon with such requirements. Consequently, the 'causality' in the response of solid materials to an external electric field is connected with the fact that the two components of the complex dielectric function are Hilbert transforms of each other, that is, the well-known Kramers–Kronig (KK) bidirectional relations. It has also been established that the dispersive optical constants are the limit on the real axis of a complex analytic function. Therefore, this function is infinitely differentiable in a neighborhood of each point in the upper complex half-plane, $\mathbb{C}^+$. The latter fundamental property implies extending the spectral variable photon energy, $E$, which is initially a real number, to the complex number $z = E + i\Gamma \in \mathbb{C}^+$, being $\Gamma$ a positive real number. An application of extending photon energies to $\mathbb{C}^+$, i.e., to complex photon energies, represents a novel procedure to transform non-analytic optical models into analytic or holomorphic [1,2]. It consists of carrying out the convolution of such a non-analytic original model with a Lorentzian-type function, which results in a spectral smoothening of the optical constants. We will return to this procedure later.

Various dispersion models have been proposed over the years for the group of semiconductor materials in order to parameterize their optical constants. These include the models of Forouhi and Bloomer [3], Campi and Coriasso [4], Tauc–Lorentz [5–8] (TL), Cody–Lorentz–Urbach [9], and Tauc–Lorentz–Urbach [10] (TLU). In the TL model, the imaginary part of the complex dielectric constant, $\epsilon_2$, becomes zero below the bandgap, whereas the experimental evidence shows otherwise. Semiconductors are materials that experience an exponential decrease in their absorption coefficient, $\alpha(= 4\pi k/\lambda)$, below the bandgap. It is called either the Urbach–Martienssen tail or simply the Urbach tail.

The TLU model was developed by Foldyna et al. [10] in order to include the Urbach band-tail to the TL model proposed by Jellison and Modine [7]. Nevertheless, this modified TLU model continues with the inappropriate use of piecewise functions. Therefore, the lack of analyticity in the TL model is still present in the TLU model. The lack of analyticity in an optical function, results in the fact that it does not have the quality to account for the optical properties of solid material, thus giving rise to inaccuracies.

In the present paper, we focus on the non-analytic TLU (NTLU) model proposed by Foldyna et al. [10], a popular model often found in the literature. Moreover, two recent papers [1,2] have put forward the procedure mentioned above, which transforms the optical constant into analytic or holomorphic and self-consistent models. This procedure suggested by Rodríguez de Marcos and Larruquert [1,2] employs the dispersion and/or the absorption term, belonging to the non-analytic model, as a 'weight function' of $(E' - E - i\Gamma)^{-1}$-type functions, and they are integrated over the complete spectral domain. Notably, Rodríguez de Marcos and Larruquert used their procedure in order to transform the non-analytic TLU model by Foldyna et al. into an analytic TLU (ATLU) model.

The primary goal of the present work is to use the analytic TLU model in order to extract the optical constants of amorphous semiconductors. To the best of our knowledge, this is the first time this approach has been utilized. What had been done so far [1] was to fit this analytic model to the experimental data belonging to $Si_3N_4$. They are available in Palik's Handbook of Optical Constants of Solids [11] and were presented by Philipp. In this work, we will determine the optical properties and the geometrical parameters, such as the average thickness and thickness variation (see the illustration regarding the wedge-shaped geometry of a thin film in Figure 1a), and of thin films of non-hydrogenated amorphous silicon (*a*-Si). The films were grown by radio-frequency magnetron-sputtering (RFMS) deposition onto transparent glass substrates at room temperature and 325 °C.

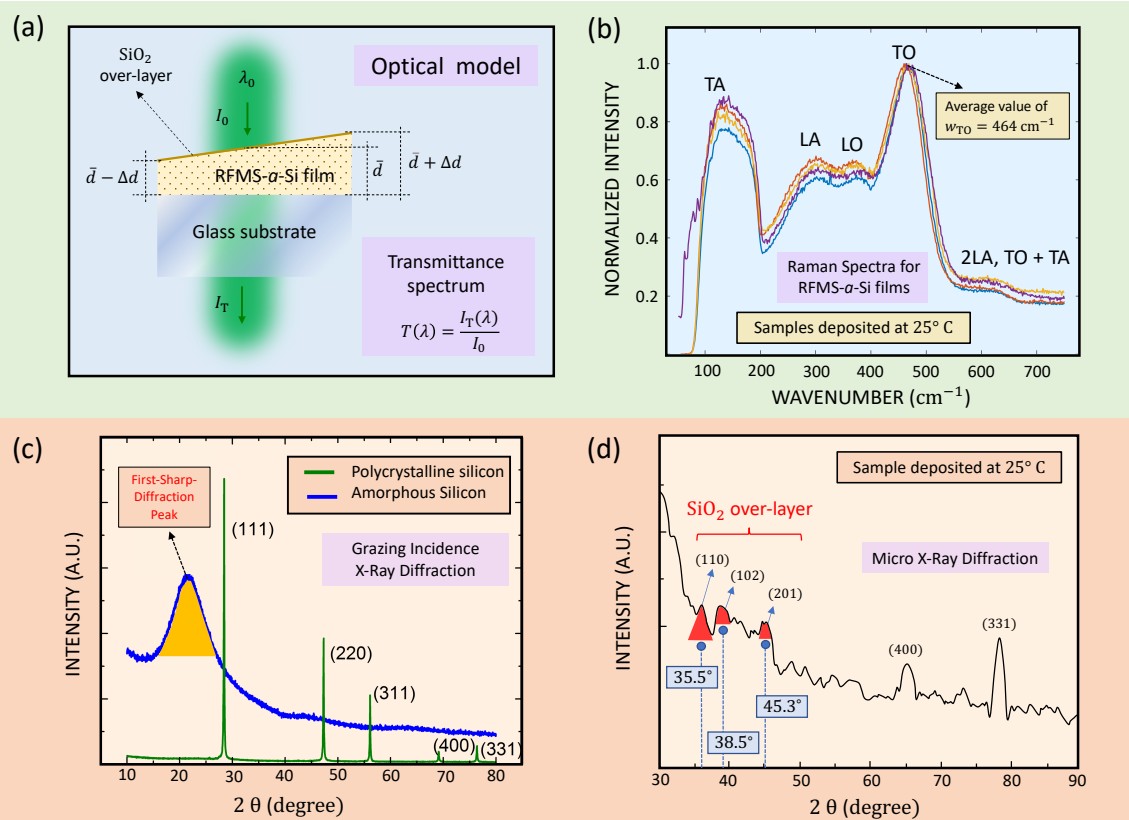

**Figure 1.** (**a**) A schematic illustration of the wedge-shaped geometry of a thin layer. (**b**) Raman spectra of four *a*-Si specimens deposited at 25 °C [12,13]. (**c**) Grazing-incidence XRD diagram and (**d**) micro-XRD pattern of an *a*-Si thin-film sample, grown at a temperature of 25 °C. In the micro-XRD diagram of this particular sample, several pronounced diffraction peaks can be noticed, which can be undoubtedly attributed to SiO$_2$ peaks (surface native oxide). The two diffraction peaks, (400) and (331), observed exclusively in this particular specimen, can correspond to crystalline planes of silicon indicative of a possible ordered nano-structure (nano-crystalline nature), embedded in the whole and absolutely *dominant* amorphous matrix (it has been considered *negligible* from the viewpoint of the assumed optical model) [14,15].

The reason behind choosing this particular material for a case study was that *a*-Si is of fundamental interest as an archetypal tetrahedral covalent amorphous network and a significant technological material for thin-film electronics [16,17]. In addition, the chosen pure *a*-Si is an excellent model amorphous semiconductor material that can be prepared reproducibly by multiple methods [18]. It should be added that new density-modulated multilayer *a*-Si thin-film anodes have been recently developed, which can be employed as a robust, high-capacity electrode for Li-ion batteries [19]. Their results have shown that these *a*-Si films can provide a very high Coulombic efficiency of up to 99% and a reversible specific capacity as high as ≈1700 mAhg$^{-1}$ after 50 cycles. These promising results can lead to *a*-Si thin-film anode materials with superior capacity and mechanical stability compared to conventional Si anodes.

## 2. Materials and Methods

The unhydrogenated *a*-Si thin layers were grown onto reasonably transparent glass (Corning Eagle XG® and 3-mm-thick Borofloat® 33) substrates using a commercial single-chamber magnetron-sputtering MV System, with a vertically adjustable cathode operated by RF power. The target–substrate distance was set to an appropriate value of 6.1 cm. The 3-inch diameter *p*-Si target (Si) was manufactured by Lesker Company (St Leonards-on-Sea,

East Sussex, UK), with a purity of 99.999%. It had an electrical resistivity of 0.005–0.020 $\Omega$cm and a mass density of 2.32 g/cm$^3$.

A K-type reference thermocouple carried out the measurement of the glass substrate temperature. Before loading the glass substrates into the deposition chamber, they were carefully cleaned via ultrasonic baths, rinsed next in deionized water, and dried finally by blown nitrogen. Before the sputtering-deposition process, the base pressure inside the chamber was approximately $3 \times 10^{-4}$ Pa. The *a*-Si thin films under study were prepared by employing two different RF powers, 525 and 450 W, associated with the deposition temperatures of room temperature and 325 °C, respectively. The Ar-working-gas pressure was between 0.7 and 4.5 Pa, corresponding to an Ar-gas flux between 17 and 70 sccm. We carefully selected these particular growth conditions in order to achieve the highest possible film-deposition rate permitted by the sputtering system used in this investigation. We also looked to produce *a*-Si thin layers, as clearly predicted by the Thornton model [20].

The structural properties and morphology of RF-sputtered *a*-Si thin films were examined via Raman spectroscopy, X-ray diffraction (XRD), and atomic force microscopy. Micro-X-ray diffraction also allowed us to perform a more profound structural and morphological analysis in order to gain more valuable information about the microstructural nature of the present sputtered *a*-Si layers. The Raman spectra were measured by using a Micro-Raman LabRam HR Evolution (Tulln, Austria) with a 633 nm He-Ne laser excitation source. The Raman spectra belonging to four representative *a*-Si samples, deposited at room temperature, are displayed in Figure 1b. The average value of the Raman shift of the TO peak, $w_{TO}$, is approximately 464 cm$^{-1}$.

XRD diagrams were collected by employing a PANalytical X'Pert Pro diffractometer, operating in a $\theta - 2\theta$ configuration with a CuK$_\alpha$ radiation (45 kV–40 mA), within the angular range of $20° < 2\theta < 80°$. Phase identification was carried out by comparison with the Inorganic Crystal Structure Database. The grazing-incidence X-ray diffraction pattern of an *a*-Si thin-film specimen grown at temperature of 25°C, is shown in Figure 1c. In this figure, the corresponding *first-sharp-diffraction peak* can be clearly noticed. Furthermore, the micro-XRD diagram of the same particular sample, exhibiting a few pronounced diffraction peaks, which can be unambiguously linked to SiO$_2$ peaks, i.e., the *surface native oxide*, is presented in Figure 1d.

The surface morphology was examined by using a standard AFM Multimode Nanoscope III-A SPM from Veeco-Digital Instruments (Cambridgeshire, UK) operating in tapping mode. The surface roughness was quantified by the root-mean-square deviation of the AFM-measured height from the mean data phase in the $5 \times 5$ μm$^2$ and $1 \times 1$ μm$^2$ images, respectively. The measured value of the surface roughness parameter $R_{q,AFM}$, for all the studied *a*-Si samples are listed later in the article. The values of the root-mean-square roughness obtained from the AFM images are in the range 0.60–2.62 nm. We consider that the present samples will give the *true* bulk dielectric function of *a*-Si. Furthermore, this is because they have nearly flat sample surface, as has been clearly demonstrated by ex situ AFM microscopy.

Finally, the normal-incidence transmittance spectra of the as-deposited *a*-Si thin-layer samples were measured at room temperature by using a Perkin-Elmer Lambda-1050 and an Agilent Cary 5000, UV/Visible/NIR double-beam spectrophotometer (Waltham, MA, USA). The geometrical and optical parameters were accurately calculated from these specific spectrophotometric data, as explained below. The optically determined layer thickness was systematically cross-checked by scanning electron microscopy (SEM: FEI Nova NanoSEM 450). The layer thickness was also independently measured for some of our *a*-Si specimens, with a Veeco Dektak 150 mechanical surface profiler.

### 3. Transforming the NTLU Model to the ATLU Model

For many semiconductors and dielectrics, the absorption coefficient $\alpha = 4\pi k/\lambda$ decreases exponentially below the optical gap, $E_g$, obeying the Urbach rule:

$$\alpha(E) = \alpha_0 \exp\left(\frac{E - E_{\text{focus}}}{E_u}\right), \tag{1}$$

where $\alpha_0$ and $E_{\text{focus}}$ define the Urbach focus, and $E_u$ representing the exponential steepness, is called the *Urbach energy*.

Foldyna et al. [10] completed the TL model by including the Urbach band-tail. Their new model, referred to as the TLU model, is given as a piecewise function with two parts for $\epsilon_2$:

$$\epsilon_{2,\text{NTLU}}\left(E; A_{\text{mp}}, E_0, E_g, C_{\text{br}}, E_c\right) = \begin{cases} \dfrac{A_u}{E} \exp\left(\dfrac{E}{E_u}\right) & \text{at } E \leq E_c, \\[4mm] \dfrac{(E - E_g)^2 A_{\text{mp}} E_0 C_{\text{br}}}{E\left(\left(E^2 - E_0^2\right)^2 + C_{\text{br}}^2 E^2\right)} & \text{at } E > E_c, \end{cases} \tag{2}$$

where $E_c$ is the connection energy between the two parts. In order to guarantee the continuity of $\epsilon_2$ and its first derivative, the parameters $A_u$ and $E_u$ must be:

$$A_u = \frac{A_{\text{mp}} E_0 C_{\text{br}} (E_c - E_g)^2 \exp(-E_c/E_u)}{E_c^4 + (C_{\text{br}}^2 - 2E_0^2)E_c^2 + E_0^4}, \tag{3}$$

$$E_u = \frac{(E_c - E_g)\left(C_{\text{br}}^2 E_c^2 + (E_0^2 - E_c^2)^2\right)}{2\left[(E_0^4 - E_c^4) + E_c E_g\left(C_{\text{br}}^2 - 2(E_0^2 - E_c^2)\right)\right]}. \tag{4}$$

In the formula describing the 'Urbach rule', the absorption coefficient $\alpha$, depends exponentially on the photon energy. However, Foldyna et al. [10] show this dependence of $\alpha$ on photon energy by expression of $\epsilon_2$, considering the fact that $n$ can be reasonably assumed to be a constant over the Urbach-band-tail fitting range. This is expressed mathematically by:

$$\begin{aligned} \epsilon_2(E \leq E_c) &= 2n(E)k(E) \approx 2n_{\text{Urbach}} k(E) = \frac{n_{\text{Urbach}} hc\alpha(E)}{2\pi E} \\ &= \frac{n_{\text{Urbach}} hc\alpha_0}{2\pi E} \exp\left(\frac{E}{E_u}\right) \exp\left(-\frac{E_{\text{focus}}}{E_u}\right) = \frac{A_u}{E} \exp\left(\frac{E}{E_u}\right), \end{aligned} \tag{5}$$

where $h$ represents Planck's constant and $c$ the speed of light in vacuum.

The TLU model presents all the drawbacks of the TL model [7], except that $\epsilon_2$ does not become zero for photon energies lower than $E_g$ in the TLU model. The disadvantages of the TLU model are the following: (*i*) the model is not analytic at $E = E_c$; (*ii*) $\epsilon_2$ is not an odd function in the exponential part (corresponding to the Urbach tail); (*iii*) we can derive $\epsilon_1$ by the KK integration, but it needs the addition of an extra parameter; (*iv*) the TLU model diverges at zero photon energy due to the term $E$ in the denominator.

In order to overcome all these drawbacks, we now make use of the integral transformation proposed by Larruquert and Rodriguez de Marcos [1,2], which has the following form:

$$\tilde{\epsilon}_{\text{Analitic}}(E) = 1 + \frac{1}{\pi} \int_{\mathbb{R}} \frac{\epsilon_{2,\text{Corrected}}(E')}{E' - E - i\Gamma} dE'. \tag{6}$$

Equation (6) uses the imaginary part of the piecewise (corrected) complex dielectric constant, $\epsilon_{2,\text{Corrected}}$, in order to find the analytic complex dielectric constant, $\tilde{\epsilon}_{\text{Analitic}}$. However, before applying Equation (6), we next present a new, corrected weight function to be inserted in the previous integral transformation, Equation (6). The expression of the

improved non-analytic TLU model suggested by Rodríguez de Marcos and Larruquert is as follows:

$$
\epsilon_{2,\text{CTLU}}\left(E; A_{\text{mp}}, E_0, E_{\text{g}}, C_{\text{br}}, E_{\text{c}}\right) = \begin{cases} A_{\text{u}} E \exp\left(\dfrac{E}{E_{\text{u}}}\right) & \text{at } E \le E_{\text{c}}, \\[2ex] \dfrac{(E - E_{\text{g}})^2 A_{\text{mp}} E_0 C_{\text{br}}}{E\left((E^2 - E_0^2)^2 + C_{\text{br}}^2 E^2\right)} & \text{at } E > E_{\text{c}}. \end{cases}
\tag{7}
$$

It has to be pointed out that this expression is certainly similar to the TLU model proposed by Foldyna et al. [10]. However, according to the uncorrected TLU model, in the limit when $E \to 0$, the function $\epsilon_2$ diverges; on the contrary, in the improved TLU piecewise function, $\epsilon_2$ more reasonably approaches zero, as the photon energy approaches zero. On the other hand, for values of the photon energy near the optical gap, $E_{\text{g}}$, the exponential term plays a very dominant role. Lastly, it must be noted that the two Urbach-tail parameters, $A_{\text{u}}$ and $E_{\text{u}}$, respectively, are chosen in such a fashion that the continuity of the new weight function and its corresponding first derivative are both guaranteed [1,2] at the particular value of the photon energy, $E = E_{\text{c}}$, which are now expressed as follows:

$$
A_{\text{u}} = \frac{A_{\text{mp}} E_0 C_{\text{br}} (E_{\text{c}} - E_{\text{g}})^2 \exp(-E_{\text{c}}/E_{\text{u}})}{E_{\text{c}}^2 [E_{\text{c}}^4 + (C_{\text{br}}^2 - 2E_0^2)E_{\text{c}}^2 + E_0^4]},
\tag{8}
$$

$$
E_{\text{u}} = \frac{E_{\text{c}}(E_{\text{c}} - E_{\text{g}})\left(C_{\text{br}}^2 E_{\text{c}}^2 + (E_0^2 - E_{\text{c}}^2)^2\right)}{2[E_{\text{g}}(E_0^4 + 3E_{\text{c}}^4) + E_{\text{c}}^2(2E_{\text{g}} - E_{\text{c}})(C_{\text{br}}^2 - 2E_0^2) - 2E_{\text{c}}^5]}.
\tag{9}
$$

The non-analytic TLU model in Equation (7) is next transformed into an analytic model. Therefore, we will perform the integration of the $(E' - E - i\Gamma)^{-1}$-type functions, considering the Urbach-tail term of Equation (7) in its spectral range, below $E_{\text{c}}$, and the TL term of Equation (7) for photon-energy values above $E_{\text{c}}$. The new analytic TLU model is found by the sum of the two already mentioned contributions:

$$
\tilde{\epsilon}_{\text{ATLU}}(E) = \tilde{\epsilon}_{\text{AU}}(E; A_{\text{mp}}, E_0, E_{\text{g}}, C_{\text{br}}, \Gamma, E_{\text{c}}) + \tilde{\epsilon}_{\text{ATL}}(E; A_{\text{mp}}, E_0, E_{\text{g}}, C_{\text{br}}, \Gamma, E_{\text{c}}).
\tag{10}
$$

The parameter $E_{\text{c}}$ must be included in the list of the ATLU free fitting parameters. Thus, this new analytic model contains *six* free parameters to be fixed, the same number as Foldyna's model. The left-hand side of Equation (10) is found from Equation (6) and is given by:

$$
\tilde{\epsilon}_{\text{AU}}(E; A_{\text{mp}}, E_0, E_{\text{g}}, C_{\text{br}}, \Gamma, E_{\text{c}}) = 1 + \frac{1}{\pi} \int_{\mathbb{R}} \frac{\epsilon_{2,\text{CU}}(E')}{E' - E - i\Gamma} dE'
\tag{11}
$$

$$
= \frac{A_{\text{u}}}{\pi} \left\{ E_{\text{complex}} \exp\left(\frac{E_{\text{complex}}}{E_{\text{u}}}\right) \left[\text{Ei}\left(\frac{E_{\text{c}} - E_{\text{complex}}}{E_{\text{u}}}\right) - \text{Ei}\left(\frac{-E_{\text{complex}}}{E_{\text{u}}}\right)\right] \right.
$$
$$
\left. + E_{\text{complex}} \exp\left(\frac{-E_{\text{complex}}}{E_{\text{u}}}\right) \left[\text{Ei}\left(\frac{E_{\text{complex}}}{E_{\text{u}}}\right) - \text{Ei}\left(\frac{E_{\text{c}} + E_{\text{complex}}}{E_{\text{u}}}\right)\right] + 2E_{\text{u}}\exp\left(\frac{E_{\text{c}}}{E_{\text{u}}}\right) - 2E_{\text{u}} \right\},
$$

with the complex photon energy expressed as:

$$
E_{\text{complex}} = E + i\Gamma,
\tag{12}
$$

and where the sign of $\epsilon_{2,\text{CU}}$ corresponding to Equation (7) was significantly reversed for negative photon-energy values to transform it into the necessary odd function. In the term $\tilde{\epsilon}_{\text{AU}}$, Equation (11), $\text{Ei}(z)$ stands for the exponential integral, a special function on the complex plane, whose expression is given by:

$$
\text{Ei}(z) = -\int_{[-z, +\infty)} \frac{e^{-t}}{t} dt = \int_{(-\infty, z]} \frac{e^t}{t} dt.
\tag{13}
$$

The function $\mathrm{Ei}(z)$ is included in the Python general-purpose programming language. Furthermore, its specific function name is 'scipy.special.expi'.

The other contribution on the right-hand side of Equation (10) is given by the expression:

$$
\begin{aligned}
\tilde{\epsilon}_{\mathrm{ATL}}(E; A_{\mathrm{mp}}, E_0, E_{\mathrm{g}}, C_{\mathrm{br}}, \Gamma, E_{\mathrm{c}}) &= 1 + \frac{1}{\pi} \int_{\mathbb{R}} \frac{\epsilon_{2,\mathrm{CTL}}(E')}{E' - E - i\Gamma} dE' = 1 + \frac{A_{\mathrm{mp}} E_0 C_{\mathrm{br}}}{\pi} \\
&\times \Big[ \mathrm{Fun}(E_{\mathrm{complex}}, \delta, \delta^*) + \mathrm{Fun}(\delta, \delta^*, E_{\mathrm{complex}}) + \mathrm{Fun}(\delta^*, E_{\mathrm{complex}}, \delta) \Big],
\end{aligned}
\tag{14}
$$

where the newly introduced function, 'Fun', is written as:

$$
\mathrm{Fun}(\alpha, \beta, \gamma) = \frac{(E_{\mathrm{g}} + \alpha)^2 \log(E_{\mathrm{c}} + \alpha) - (E_{\mathrm{g}} - \alpha)^2 \log(E_{\mathrm{c}} - \alpha)}{\alpha(\alpha^2 - \beta^2)(\alpha^2 - \gamma^2)},
\tag{15}
$$

and the two parameters $\delta$ and $\delta^*$ appearing in Equation (14), are given by the respective expressions:

$$
\delta = \sqrt{E_0^2 - \left(\frac{C_{\mathrm{br}}}{2}\right)^2} - i\frac{C_{\mathrm{br}}}{2},
\tag{16}
$$

and

$$
\delta^* = \sqrt{E_0^2 - \left(\frac{C_{\mathrm{br}}}{2}\right)^2} + i\frac{C_{\mathrm{br}}}{2}.
\tag{17}
$$

In the case of the previous analytic TLU model, we can extract the complex refractive index, $\tilde{N} = n + ik$, directly from Equation (10) considering the essential relationship $\tilde{\epsilon} = \tilde{N}^2$. From this fundamental equation are derived the following relations:

$$
\begin{aligned}
\epsilon_1 &= n^2 - k^2, \\
\epsilon_2 &= 2nk,
\end{aligned}
\tag{18}
$$

or alternatively,

$$
\begin{aligned}
n &= \sqrt{\frac{1}{2}\left(\sqrt{\epsilon_1^2 + \epsilon_2^2} + \epsilon_1\right)}, \\
k &= \sqrt{\frac{1}{2}\left(\sqrt{\epsilon_1^2 + \epsilon_2^2} - \epsilon_1\right)}.
\end{aligned}
\tag{19}
$$

Thus, we can fit the free parameters of the ATLU model to the collected experimental data.

## 4. Fitting the Universal Transmission Formula to the As-Measured Spectrum

The universal optical-transmission formulae for normal-incidence, $T_{\Delta d}(\lambda)$, employed in this investigation, is a new equation recently published by some of the authors of this work [21–23]. This transmission expression is also valid for highly non-uniform (strongly wedge-shaped) thin layers deposited onto thick transparent glass substrates, and it is as follows:

$$
\begin{aligned}
&T_{\Delta d}(\lambda; n(\lambda), k(\lambda), s(\lambda), \bar{d}, \Delta d) \\
&= \frac{A\left[\left(\tan^{-1}\left(\frac{C}{D}\right) - \tan^{-1}\left(\frac{B}{D}\right)\right) + \pi(N_{\mathrm{c},2} - N_{\mathrm{c},1})\right]}{D(\psi_2 - \psi_1)} x_{\mathrm{ave}},
\end{aligned}
\tag{20}
$$

where $x_{\mathrm{ave}} = \exp(-\alpha\bar{d})$, $\alpha = 4\pi k/\lambda$, $\psi_1 = 4\pi n(\bar{d} - \Delta d)/\lambda$, $\psi_2 = 4\pi n(\bar{d} + \Delta d)/\lambda$ and, additionally, the rest of the parameters used in the expression of transmission are the following:

$$
\begin{aligned}
A &= 32(n^2 + k^2)s, \\
B &= x_{\text{ave}}(F + E(G + H\,x_{\text{ave}}))\tan(\psi_1/2), \\
C &= x_{\text{ave}}(F + E(G + H\,x_{\text{ave}}))\tan(\psi_2/2), \\
D &= \sqrt{E^2 - x_{\text{ave}}^2(F^2 + G^2 - 2EH - H^2\,x_{\text{ave}}^2)}, \\
E &= \left((n+1)^2 + k^2\right)\left((n+1)\left(n + s^2\right) + k^2\right), \\
F &= 2k\left(2(n^2 + k^2 - s^2) + (n^2 + k^2 - 1)(s^2 + 1)\right), \\
G &= 2\left((n^2 + k^2 - 1)(n^2 + k^2 - s^2) - 2k^2(s^2 + 1)\right)\left((n-1)(n - s^2) + k^2\right), \\
H &= (n-1)^2 + k^2.
\end{aligned}
\tag{21}
$$

Finally, the mathematical expressions for the two proposed correcting integers, $N_{\text{c},1}$ and $N_{\text{c},2}$, respectively, are the following:

$$
N_{\text{c},1} = \text{round}(\psi_1/2\pi),
\tag{22}
$$

and

$$
N_{\text{c},2} = \text{round}(\psi_2/2\pi).
\tag{23}
$$

The new function, 'round', does round off its arguments to its closest exact integers. The novel universal expression for the transmission, Equation (20), is a continuous function employed to carry out the comprehensive optical characterization of a wide variety of thin films of amorphous semiconductors with excellent accuracy. We can successfully eliminate the current limiting value of the non-uniformity or wedging geometrical parameter, $\Delta d_{\text{max}} = \lambda/4n(\lambda)$ (see details in [23,24]). Therefore, we can now calculate the optical constants of strongly wedge-shaped layers. Thus, introducing the new two angular correcting parameters, $N_{\text{c},1}$ and $N_{\text{c},2}$, respectively, in the proposed universal formula for the normal-incidence transmission of a real thin layer is crucial to eliminating the previous constraint, $\Delta d < \Delta d_{\text{max}}$.

Normal-incidence transmission is, as already mentioned, a function of $\bar{d}$, $\Delta d$, $n(\lambda)$ and $\alpha(\lambda)$. In addition, we have initially calculated the values of the refractive index of the glass substrate alone, $s(\lambda)$, where the RFMS-Si layer was deposited. When we adopt the analytic ATLU model in the present study, the theoretical normal-incidence transmission formula, Equation (20), will depend upon *eight* free model parameters: $\bar{d}$, $\Delta d$, $A_{\text{mp}}$, $E_0$, $C_{\text{br}}$, $\Gamma$, and $E_{\text{c}}$. In the following section, we will accurately calculate those eight free parameters by direct computer fitting of the above expression for $T_{\Delta d}(\lambda)$ to the experimental transmission spectrum, with the help of a devised ad hoc comprehensive computer program called OCISPY.

This program uses the so-called inverse synthesis method (or reverse engineering) [25,26], as an alternative to the Swanepoel transmission-envelope procedure, which exclusively employs the Fabry–Pérot (FP) interference fringes. Our approach has the clear advantage of being able to apply it to very thin layers with average thicknesses below 100 nm. In that particular case, the number of FP interference fringes is relatively low or practically non-existent.

To illustrate the dispersion-model-fitting method, we have analyzed a thin film of hydrogen-free *a*-Si in-depth that presents a large number of FP fringes in the whole measured spectral range, with the values of $\lambda > 700$ nm. It must be emphasized that there is a weak optical absorption in the measured spectral range. This non-negligible absorption is the universal characteristic feature of the absorption spectra near the band edges, found in both crystalline and non-crystalline semiconductors. As previously mentioned, the Urbach–Martienssen exponential absorption edge models the absorption in that region.

The absorption coefficient, $\alpha$, verifies the general Urbach rule expressed in Equation (1) near the band edges when the optical-induced electrons transit from the valance band up to conduction-energy-band tails.

### 5. OCISPY (Optical Characterization by Inverse Synthesis): Python-Coded Computer Program for Determining the Optical Properties of Amorphous Semiconductor Films by Employing Multiple Dispersion Models

We built a Python-based computer program named OCISPY in order to perform the accurate and comprehensive optical characterization of both uniform and non-uniform amorphous semiconductor thin layers. As previously indicated, it belongs to the category of reverse-synthesis approach. A very simplified flowchart of its algorithm is shown in Figure 2. This program permits the very fast fitting of a model-simulated normal-incidence transmittance spectrum to the as-measured transmittance spectrum of the amorphous semiconducting layer under study by fitting eight free model parameters. *Two* of them correspond to the geometrical parameters, namely, the average film thickness and, very interestingly, the wedging parameter, $\bar{d}$ and $\Delta d$, respectively. Furthermore, the remaining *six* fitting parameters belong obviously to the adopted analytic TLU model.

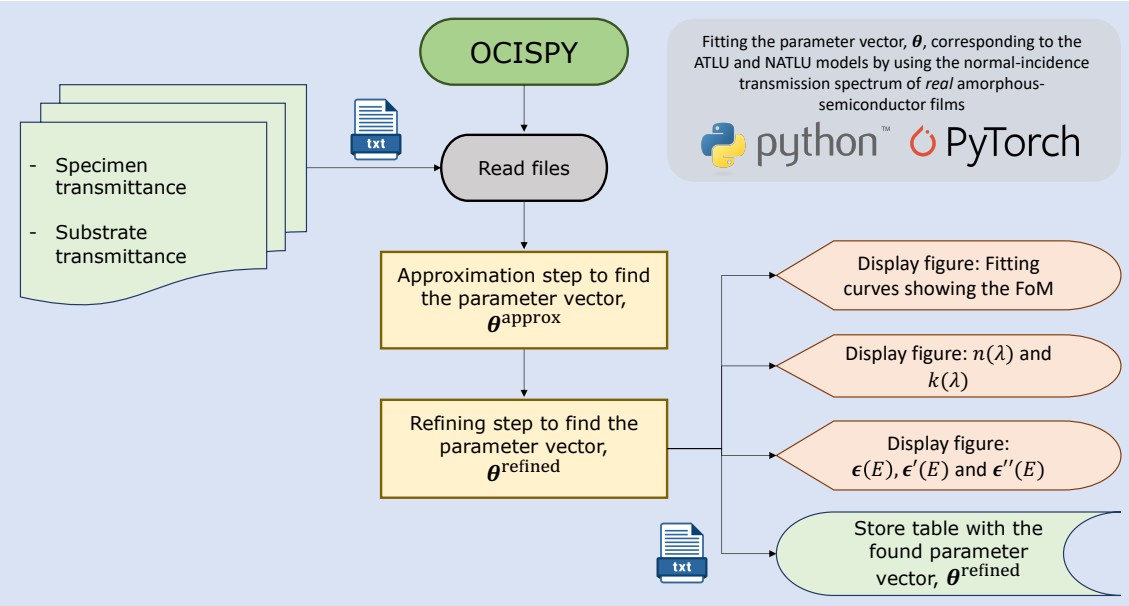

**Figure 2.** A very simplified flowchart corresponding to the optimization algorithm for the comprehensive optical characterization of amorphous-semiconductor thin layers. This specific algorithm is implemented in the Python-based computer program, OCISPY, presented and employed in this work. $\theta$ stands for the *parameter vector* introduced in the program. It very accurately solves the problem of extracting the optical properties, together with the average layer thickness, $\bar{d}$, and the wedging parameter, $\Delta d$, of non-crystalline-semiconductor thin films.

The fundamental idea behind the computer program OCISPY in order to accurately calculate all the free fitting parameters is to minimize the chosen figure-of-merit (FoM) or cost function:

$$\text{FoM} = 100 \times \text{RMSE} = 100 \times \sqrt{\frac{\sum_{i=1}^{N}(T_{i,\text{meas}} - T_{i,\text{simu}})^2}{N_{\text{exp}}}}, \tag{24}$$

where $N_{\text{exp}}$ stands for the total number of experimentally measured values of transmittance, $T_{i,\text{meas}}$ denotes each particular as-measured transmittance value, and $T_{i,\text{simu}}$ stands for each respective simulated transmittance value. These transmittance values correspond to all those measured wavelengths for which the glass substrate is quasi-transparent. The FoM

function measures the root-mean-square error (RMSE) in percentage, corresponding to the existing differences between the experimental and generated transmission values. It must be added that a two-step optimization algorithm carries out the fitting process. The first step corresponds to an approximation phase, using a global optimization method (random search). In the second step, we refine the results by using and comparing the results from the quasi-Newton algorithm and the heuristic algorithm devised by Nelder-Mead (downhill or simplex algorithm).

## 6. Extracting the Optical Constants *n* and *k* by Employing the Original NTLU and the Novel ATLU Dispersion Model

We have employed the proposed optical-characterization method based on the NTLU (original) and ATLU (new) optical-dispersion models. The *eight* free-fitting parameters $\bar{d}$, $\Delta d$, $A_{mp}$, $E_0$, $E_g$, $C_{br}$, $\Gamma$, and $E_c$, were accurately calculated along with their corresponding low values of the figure of merit (FoM). Figure 3 shows the experimental (open scatters) and the simulated fit (solid curves) of the normal-incidence transmission spectra for seven selected representative H-free RFMS-*a*-Si thin samples, which were grown either at room temperature or at 325 °C.

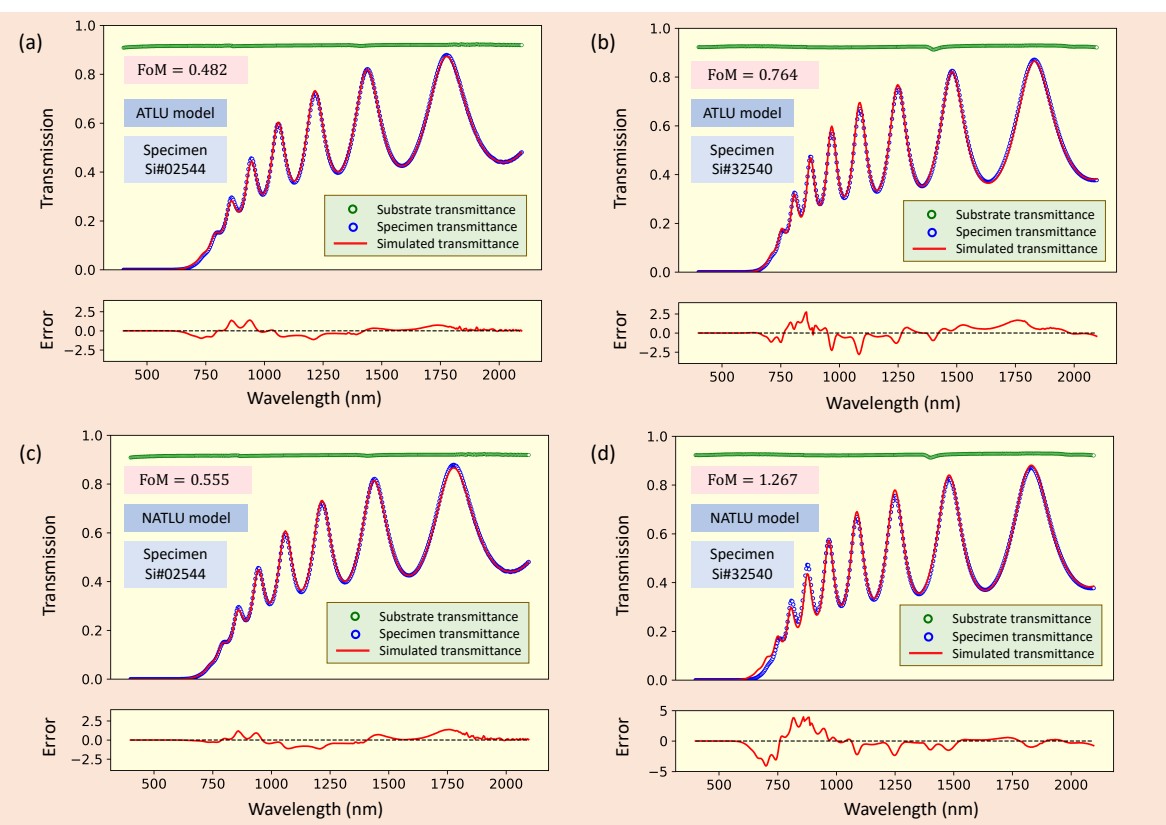

**Figure 3.** Two examples of the typical normal-incidence transmission spectrum of an approximately 1.1-μm-thick amorphous Si thin film that have been sputtered on a transparent glass substrate, held at a temperature of (**a**) 25 °C and (**b**) 325 °C, during the sputtering deposition process. Furthermore, the NATLU- and ATLU-generated transmission spectra of the previous two *a*-Si specimens, are also displayed. The relative difference between the experimental and simulated transmittance curves is also displayed in this figure for the two representative *a*-Si samples.

We summarize in Table 1 the best-fit parameters, along with the Urbach energy and amplitude, $E_u$ and $A_u$, respectively, for both dispersion models. A first point to remark on is that the fitted average film thicknesses, $\bar{d}$, are very close to the experimental values measured by the cross-sectional SEM micrographs (see Table 1 for the values of $d_{SEM}$), which partially validates the two considered TLU dispersion methods. On the other hand,

the $E_g$ values are closely comparable to those of *a*-Si reported in [20], further supporting the use of both TLU models to a certain extent. Although FoM values are reasonable for both models, one can recognize that the FoM is significantly lower for the ATLU model. This finding suggests that the spectra simulated with the ATLU model are closer to the ground-truth real transmittance curves, consistent with the analytical investigation of the novel ATLU model. In real non-simulated spectra, it is complicated to establish a threshold FoM number for good fits, as it not only depends on the model but also on the spectrum noise and the experiment's accuracy. We use the FoM values to compare the models rather than as an absolute measure of their performance.

**Table 1.** Values of all the ATLU model parameters for RFMS-*a*-Si thin layers, each prepared with a different Ar-gas pressure. In the sample identification (ID) code, the first adopted three numbers indicate the deposition temperature in degrees Celsius, whereas the next two numbers show the Ar-working pressure during the deposition in the particularly chosen unit decipascal. In addition, the values of the adopted cost or merit function, FoM, for those *a*-Si samples under study are presented. The values of the Urbach energy and amplitude, $E_u$ and $A_u$, respectively, obtained from the previous ATLU-model parameters, are also listed in the table. On the other hand, the value of the geometrical parameter $\bar{d}$ ($\equiv d$ for these *a*-Si samples), is also indicated. The best-fit values for $\Delta d$ are not shown in the table, since they are practically zero in all the roughly 1.1-μm-thick-specimens investigated; only layers with thickness about or less than 800 nm exhibited a fitted value of $\Delta d > 0$. Moreover, the experimentally determined values of the surface roughness parameter, $R_{q,AFM}$, for the seven *a*-Si specimens are also presented. In addition, the film thickness, $d_{SEM}$, measured from the cross-sectional SEM micrographs corresponding to the present *a*-Si specimens under study, are also listed.

| Sample ID | Model | FoM | $\bar{d}$ (nm) | $R_{q,AFM}$ (nm) | $A_{mp}$ (eV) | $E_0$ (eV) | $E_g$ (eV) | $C_{br}$ (eV) | $\epsilon_1(\infty)/\Gamma$ (eV) | $E_c$ (eV) | $A_u$ (eV$^{\pm 1}$) | $E_u$ (meV) |
|---|---|---|---|---|---|---|---|---|---|---|---|---|
| Si#02544 | NTLU | 0.555 | 1122 | 1.84 | 100 | 3.73 | 1.22 | 2.44 | 1.00 (fixed) | 1.75 | $1.3 \times 10^{-3}$ | 239 |
| $d_{SEM}$=1142 nm | ATLU | **0.482** | 1118 | 1.84 | 105 | 3.66 | **1.27** | 2.35 | 0.0074 | 1.85 | $4.0 \times 10^{-3}$ | **355** |
| Si#02532 | NTLU | 0.667 | 1179 | 1.73 | 110 | 3.59 | 1.19 | 2.14 | 1.00 (fixed) | 1.70 | $1.2 \times 10^{-3}$ | 230 |
| $d_{SEM}$=1205 nm | ATLU | **0.576** | 1174 | 1.73 | 112 | 3.53 | **1.22** | 1.93 | 0.0046 | 1.81 | $4.9 \times 10^{-3}$ | **352** |
| Si#02511 | NTLU | 1.073 | 1364 | 1.66 | 121 | 3.52 | 1.11 | 1.95 | 1.00 (fixed) | 1.60 | $1.2 \times 10^{-3}$ | 217 |
| $d_{SEM}$=1380 nm | ATLU | **0.818** | 1360 | 1.66 | 113 | 3.46 | **1.09** | 1.52 | $\approx 0$ (*) | 1.68 | $7.3 \times 10^{-3}$ | **356** |
| Si#32540 | NTLU | 1.267 | 1053 | 2.62 | 141 | 3.47 | 1.45 | 1.53 | 1.00 (fixed) | 2.13 | $1.7 \times 10^{-3}$ | 267 |
| $d_{SEM}$=1070 nm | ATLU | **0.764** | 1063 | 2.62 | 166 | 3.45 | **1.49** | 2.38 | 0.0353 | 1.79 | $8.9 \times 10^{-6}$ | **167** |
| Si#32527 | NTLU | 1.226 | 1035 | 1.70 | 303 | 3.01 | 1.75 | 2.39 | 1.00 (fixed) | 2.25 | $8.0 \times 10^{-4}$ | 234 |
| $d_{SEM}$=1072 nm | ATLU | **0.904** | 1044 | 1.70 | 321 | 2.89 | **1.64** | 3.62 | 0.0230 | 1.93 | $2.8 \times 10^{-5}$ | **183** |
| Si#32516 | NTLU | 1.535 | 1042 | 1.30 | 278 | 3.05 | 1.73 | 2.06 | 1.00 (fixed) | 2.24 | $5.8 \times 10^{-4}$ | 226 |
| $d_{SEM}$=1065 nm | ATLU | **0.724** | 1057 | 1.30 | 226 | 3.23 | **1.53** | 2.98 | 0.0236 | 1.76 | $5.9 \times 10^{-7}$ | **130** |
| Si#32507 | NTLU | 1.153 | 1197 | 0.60 | 250 | 3.17 | 1.75 | 1.65 | 1.00 (fixed) | 2.44 | $2.3 \times 10^{-3}$ | 270 |
| $d_{SEM}$=1247 nm | ATLU | **0.823** | 1206 | 0.60 | 171 | 3.43 | **1.44** | 2.04 | 0.0335 | 1.78 | $3.7 \times 10^{-5}$ | **187** |

* The best-fit value of $\Gamma$ in this particular case is extremely close to zero.

We will then commence analyzing both the NTLU- and ATLU-fitted results. First of all, the variation range of the amplitude parameter, $A_{mp}$, for the particular case of the three *a*-Si samples under analysis, grown at the temperature of 25 °C, and corresponding to the adoption of the ATLU model, was found to range from 105 to 113 eV, increasing with the decrease in Ar-gas pressure. With the adoption of the NTLU model, the values of $A_{mp}$ were also found to increase from 100 to 121 eV, again as the Ar-working pressure decreases (see Table 1). In the specific case of the four *a*-Si specimens deposited at a temperature of 325 °C, it was found with both the analytic and non-analytic TLU models, that there is no clear trend of $A_{mp}$, as the Ar pressure decreases. What is certainly true is that their values of the parameter $A_{mp}$ are much lower than those obtained for the *a*-Si samples grown on unheated glass substrates. This is the direct result of the *a*-Si material being *less dense*, as will be reconsidered later in the paper.

On the other hand, the optical-band-gap parameter, $E_g$, was found to be in the variation ranges from 1.22 to 1.11 eV for *a*-Si layers grown at room temperature, following the NTLU model, and from 1.27 to 1.09 eV according to the ATLU model, in both cases, as the Ar pressure decreases. The samples deposited at 325 °C, show a value of $E_g$, calculated from

the NTLU model, excessively large around 1.75 eV, a typical value of hydrogenated *a*-Si films, whereas the values of the parameter $E_g$ determined by using the ATLU model are lower than those obtained by its corresponding non-analytic model, yet higher than those obtained with the same ATLU dielectric function, but for the three samples grown at 25 °C. The concrete value of $E_g$ when adopting the ATLU model fluctuates around 1.50 eV, for the values of the samples deposited at 325 °C

In addition, the offset parameter, $\epsilon_1(\infty)$, has been held conveniently fixed to one in the specific case of the NTLU-model according to Foldyna et al. [10]. In contrast, the different values of the broadening parameter, $\Gamma$, for the new ATLU model are listed in Table 1. Furthermore, the Urbach energy and amplitude, $E_u$ and $A_u$, respectively, determined from the parameters of both models, are also shown in Table 1. It must be emphasized that the bandgap parameters $E_g$ of our as-deposited *a*-Si layers increase with higher deposition temperature for both dispersion models. However, the band tail $E_u$ displays decreases with higher deposition temperature in the case of the new ATLU model, while it practically shows *no change* with deposition temperature in the NTLU model. According to the ATLU model's results, the *a*-Si structure is more ordered at the higher deposition temperature of 325 °C. This behavior with increasing deposition temperature has frequently been reported in the literature, which would partially validate the new ATLU approach and certainly reject the original NTLU model.

It must be borne in mind that following Tanaka [27], all the data concerned with the Urbach rule reported so far reveal the existence of a minimal Urbach energy of approximately 50 meV, in non-metallic disordered materials. This energy can be connected with an intrinsic density fluctuation, which is inherent to nano-scale, medium-range non-crystalline structures. Other disordered structures, such as a defective structure, tend to very strongly increase the Urbach energy, as in the case of hydrogen-less *a*-Si. On the other hand, O'Leary [28,29] had previously pointed out that the Urbach parameter, $E_u$, is a clear indicator of the amount of intermediate-range order, in such a way that: the greater the topological order, the lower the value of $E_u$. O'Leary reported the value of the Urbach energy of 229 meV associated with *a*-Si grown by molecular beam epitaxy. This value of the $E_u$ parameter lies between the value of 85 meV belonging to the plasma-deposited *a*-Si and that associated with the sputter-deposited *a*-Si of 247 meV, the greatest of the three previous different *a*-Si materials.

Moreover, the lack of sensitivity to the Urbach energy $E_u$ to the deposition temperature by the NTLU model is certainly unexpected and unacceptable, as mentioned before, since they conflict with the trend of the other free-fitting parameters. Therefore, it seems more reasonable to adopt the new ATLU dielectric function as an optical dispersion model. The best-fit values of $E_u$ computed by the more reasonable ATLU function of around 354 meV and 167 meV for our *a*-Si samples, for the deposition temperatures of 25 °C and 325 °C, respectively, are undoubtedly consistent with the three aforementioned reported values. The specific value of 354 meV for a deposition temperature of 25 °C is notably the largest among all of them, thus corresponding to the largest nano-scale, structural disorder of all the unhydrogenated *a*-Si materials previously considered. Furthermore, given the whole set of results, it is plausible to infer that growth temperature is the dominant factor in influencing all optical parameters. In this fashion, the values of the Urbach energy, $E_u$, of the *a*-Si layers grown at 325 °C are about half of the those *a*-Si films sputtered at room temperature. It means that the topological disorder existing in the atomic structure of the present films deposited at higher temperature is, indeed, much less than that found on those layers coated upon unheated glass substrates. This clearly agrees with the fact that the fitted values of bandgap parameter $E_g$ are found to be smaller for the *a*-Si samples sputtered at room temperature. The parameter $E_g$ is also usually considered a measure of the amount of intermediate-range order, in this way, greater topologycal order corresponds to larger $E_g$ [28,29]. We, thus, fully demonstrate the consistency and accuracy of the novel optical characterization method based on the ATLU dispersion model compared to the original NTLU model.

We plot the calculated optical constants $(n(\lambda), k(\lambda))$ *versus* wavelength in Figure 4. It must be stressed that for most of the wavelength range, it is verified that $dn/d\lambda < 0$. This is the case for 'normal' optical dispersion. However, near the intense electronic absorption band, the derivative of $n$ is reversed, i.e., $dn/d\lambda > 0$. This expression corresponds to the so-called 'anomalous' (i.e., abnormal) dispersion region. Furthermore, note that as the wavelength continues decreasing, $n$ approaches the unity from values of $n$ less than the unity. That is, it is again verified that $dn/d\lambda < 0$. It has to be noted that from the KK dispersion relations, the behavior of the refractive index must be necessarily interrelated to the existing electronic absorption band of the present RFMS-*a*-Si thin films, clearly observed by the behavior of the extinction coefficient, $k$ (see Figure 4). In other words, it is ensured in this fashion that the complex refractive index, $n + ik$, adopts a physically plausible shape, i.e., light absorption causes the referred effect of anomalous dispersion (absorption 'bumps'/peaks in $k$ produce 'wiggles' in $n$), and, significantly, the larger the area-under-the-curve of the bump, the stronger the effect that takes place on the curve of the real refractive index, $n$.

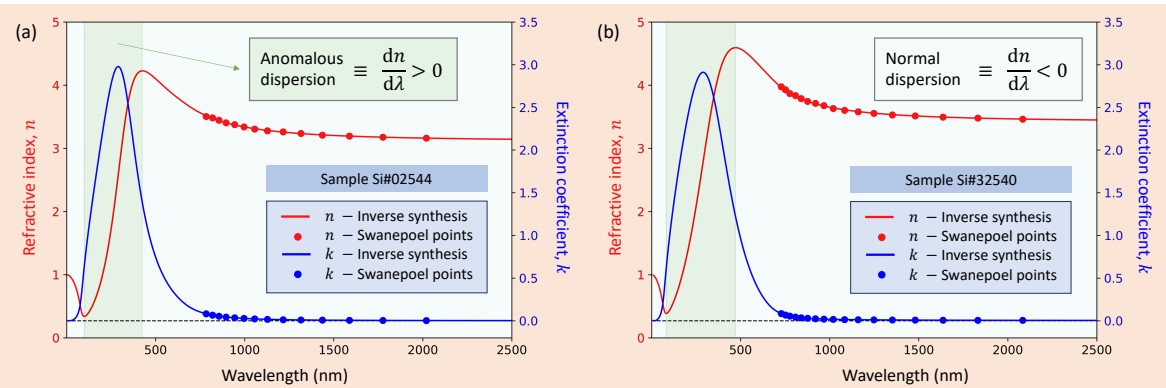

**Figure 4.** Refractive index, $n$, and extinction coefficient, $k$, *versus* wavelength, for *a*-Si thin layers (specimens (**a**) Si#02544 and (**b**) Si#32540), obtained by inverse synthesis, with the help of the holomorphic ATLU function adopted in this work. In addition, the values of $n$ and $k$ found by employing the model-free Swanepoel envelope technique are also shown for comparison in this figure.

Continuing with the analysis of the present results, the real and imaginary parts of the complex dielectric constant, $\epsilon_1$ and $\epsilon_2$, respectively, as a function of the photon energy, are displayed in Figure 5 for some representative *a*-Si specimens.

It should be noted that the broad peak of $\epsilon_2$, whose maximum value is denoted as, $\epsilon_{2,\max}$, is related to the splitting of the bonding and antibonding electronic states. This peak is located in the case of the specimen Si#02544 at a photon energy, $E(\epsilon_{2,\max})$, of approximately 3.68 eV, extremely close to the obtained value of the ATLU parameter, $E_0$, of 3.66 eV. This single smeared peak of the imaginary part of the dielectric function is typically found in tetrahedrally bounded non-crystalline semiconductors [20], as in elemental semiconductor materials such as Si and Ge.

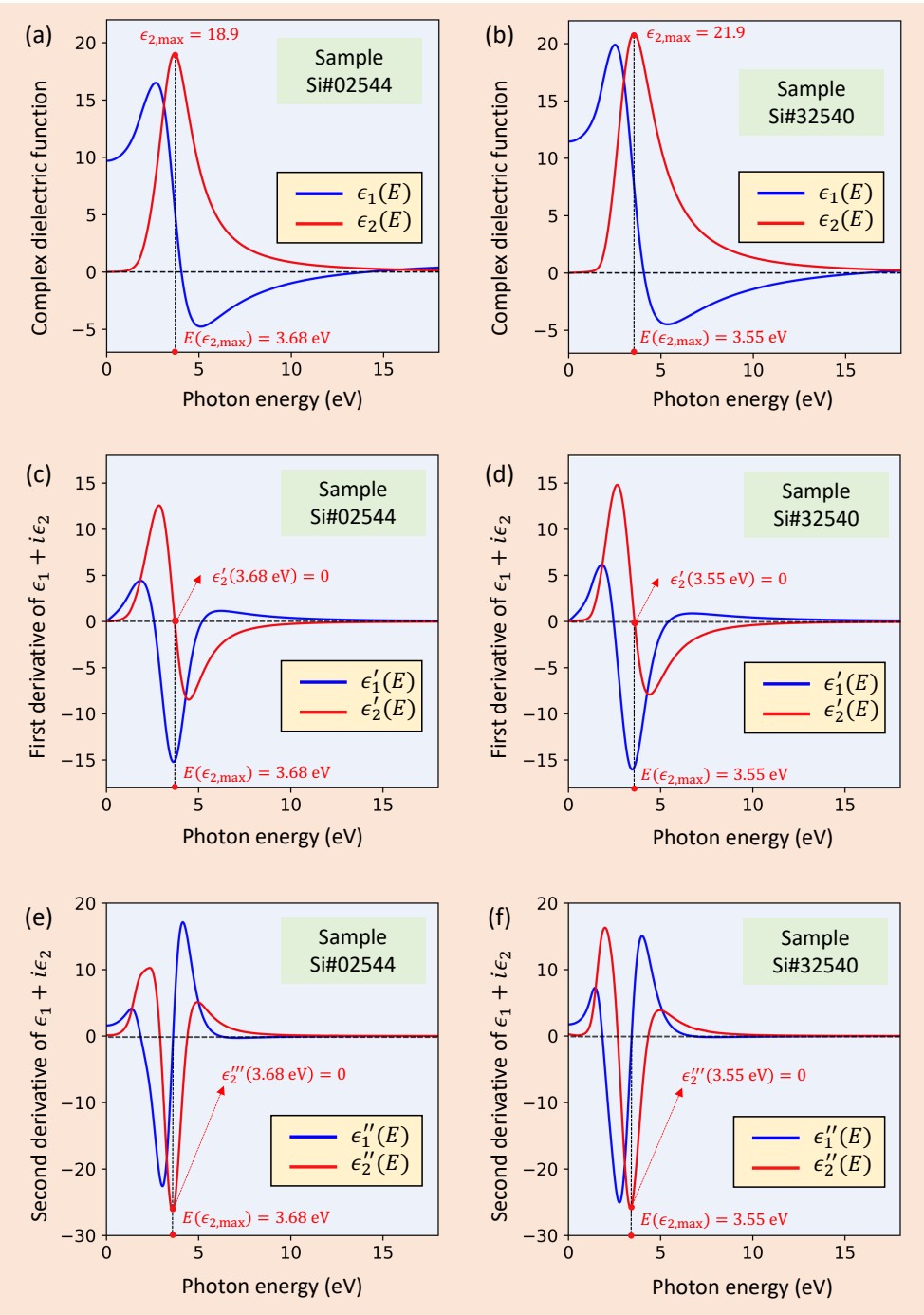

**Figure 5.** Photon-energy dependence of the real and imaginary parts of the complex dielectric constant of two *a*-Si thin-film samples: (**a**) Si#02544 and (**b**) Si#32540. Shown also in this figure, the first and second derivative of $\epsilon_1$ and $\epsilon_2$ (**c**–**f**), demonstrating that they are certainly twice differentiable, as should be the case considering that the ATLU dielectric function is complex analytic.

We see in Figure 5 that the amorphous material shows the highest value, $\epsilon_{2,\text{max}}$, which corresponds to that found for a dense hydrogen-free fully *a*-Si in self-implanted amorphous silicon. The greatest value $\epsilon_{2,\text{max}} = 26.6$ corresponds to the specific hydrogen-less *a*-Si material reported by Adachi and Mori [30,31], found at the photon energy $E(\epsilon_{2,\text{max}}) = 3.45$ eV. We have taken these concrete values as a reliable reference to compare with our results, and we have found the values of the two dielectric-constant parameters $\epsilon_{2,\text{max}}$ and $E(\epsilon_{2,\text{max}})$ of 27.4 and 3.50 eV, respectively, undoubtedly very consistent with those values reported by Adachi and Mori for the specific case of the densest *a*-Si (see Table 2). Furthermore,

the smaller values of the height of the $\epsilon_2$ peak for the rest of our RFMS-*a*-Si specimens can be reasonably accounted for by the smaller mass density, associated with the *voided structure* [20,32–34] of the studied RFMS-*a*-Si thin films. Finally, also presented in Figure 5, the first and second derivative of $\epsilon_1$ and $\epsilon_2$, unambiguously demonstrating that the complex dielectric function, $\epsilon_1 + i\epsilon_2$, is twice differentiable, as it must be expected taking into account that such a ATLU function is *holomorphic,* (that is, infinitely differentiable).

**Table 2.** The obtained values for three dielectric-constant parameters $\epsilon_1(0)$, $\epsilon_{2,\max}$, and $E(\epsilon_{2,\max})$ (see Figure 5), for the seven *a*-Si specimens under analysis, are all listed in this table. The fitted values have been found by adopting the holomorphic ATLU function.

| Parameter | Si#02544 | Si#02532 | Si#02511 | Si#32540 | Si#32527 | Si#32516 | Si#32507 |
|---|---|---|---|---|---|---|---|
| $\epsilon_1(0)$ | 9.69 | 11.1 | 13.1 | 11.5 | 12.9 | 12.8 | **12.9** |
| $\epsilon_{2,\max}$ | 18.9 | 24.7 | 34.7 (*) | 21.9 | 18.7 | 21.3 | **27.4** |
| $E(\epsilon_{2,\max})$ (eV) | 3.68 | 3.54 | 3.45 | 3.55 | 3.44 | 3.45 | **3.50** |

(*) We consider that this value of $\epsilon_{2,\max}$ is certainly overestimated (it could even be considered *unphysical*) compared with the best-fit reported values obtained from the literature. Furthermore, it is mathematically well explained by the extremely low obtained value of broadening parameter, $\Gamma$.

## 7. The Alternate Swanepoel Technique: Model-Free Determination of *n* and *α*, for Uniform Thin Films on Transparent Substrates

Our unhydrogenated *a*-Si thin-film samples present a reasonable uniform thickness (it has been found that $\Delta d \approx 0$ for all the samples investigated). Therefore, the normal-incidence transmission spectrum obtained from a spectrophotometer shows many FP interference fringes (see Figure 6). One useful and popular method that makes use of these interferences fringes to determine the optical properties of the material is the so-called *model-free Swanepoel method* [24,32,35–38], which is based on an earlier work by Manifacier et al. [39].

We should calculate the refractive index of the substrate before extracting the optical properties of the uniform thin films. For a glass substrate with negligible absorption, that is, $k \leq 0.1$ and $\alpha \leq 10^{-2}$ cm$^{-1}$, in the range of operating wavelengths, the refractive index *s* is:

$$s = \frac{1}{T_s} + \sqrt{\frac{1}{T_s^2} - 1}, \tag{25}$$

where $T_s$ is the transmission of the bare substrate. With this refractive index *s* already known, the next step is constructing the two top and bottom envelopes around the FP interference fringes in the transmission spectrum, as indicated in Figure 6.

Following the Swanepoel method, the refractive index of the uniform film, *n*, in the weakly absorbing spectral region where the absorption coefficient, $\alpha$, is $\alpha \approx 0$, is determined by using the very accurate closed-form expression [35]:

$$n = \sqrt{2s\frac{T_+ - T_-}{T_+ T_-} + \frac{s^2 + 1}{2} + \sqrt{\left(2s\frac{T_+ - T_-}{T_+ T_-} + \frac{s^2 + 1}{2}\right)^2 - s^2}}, \tag{26}$$

where $T_+$ and $T_-$ are, respectively, the envelope-curve values at the wavelengths at which the upper- and lower-envelope curves and the transmission spectrum are *tangential*, as displayed in Figure 6. The first approximate values of the refractive index of the thin-film material, $n_{\text{crude}}$'s, were calculated by using Equation (26), at the particular wavelengths corresponding to the tangency points. If $n_1$ and $n_2$ are the refractive indices at two adjacent top-envelope (or bottom-envelope) tangential points at the wavelength $\lambda_1$ and $\lambda_2$, respectively, then by employing the basic equation for the appearance of interference fringes in the normal-incidence optical transmission spectra of parallel-faced layers (in this particular layer geometry its thickness is just called, *d*),

$$2n(\lambda_{\text{tan}})d = m\lambda_{\text{tan}}, \tag{27}$$

where $m$ is the interference order number. The first value of the film thickness, $d_{\text{crude}}$, is next determined by the following expression:

$$d_{\text{crude}} = \frac{\lambda_1 \lambda_2}{2(\lambda_1 n_2 - \lambda_2 n_1)}.$$ (28)

As a result of the thin-film absorption, the tangent points $\lambda_{\text{tan}}$ from Equation (27) are not exactly the maxima and minima of the transmission spectrum, but are instead corrected at nearby particular tangent locations [24]. A list of values of $d_{\text{crude}}$ is obtained by employing Equation (28) for each pair of consecutive top-envelope (or bottom-envelope) tangential points. We can then calculate the average thickness value, $\bar{d}_{\text{crude}}$, from that list. Next, we use these first values for the average thickness and refractive index to approximate the (non-exact) interference order numbers, $m_0$'s, from Equation (27). A theoretical value $m$ is an exact integer for an upper tangency point and an exact half-integer for a lower tangency point. In addition, we find more accurate values of the thin-film thickness, $d_{\text{acc}}$'s, by rounding-off to the nearest exact integer and half-integer values of the order number, $m$. With this set of exact order numbers and with an accurate average thickness $\bar{d}_{\text{acc}}$, a new set of improved final values of the thin-layer refractive index, $n_{\text{acc}}$'s, are calculated. It must also be pointed out that the values of some FP-interferences order numbers are displayed below in the paper, in order to enrich the proposed comprehensive analysis of the normal-incidence transmission spectra.

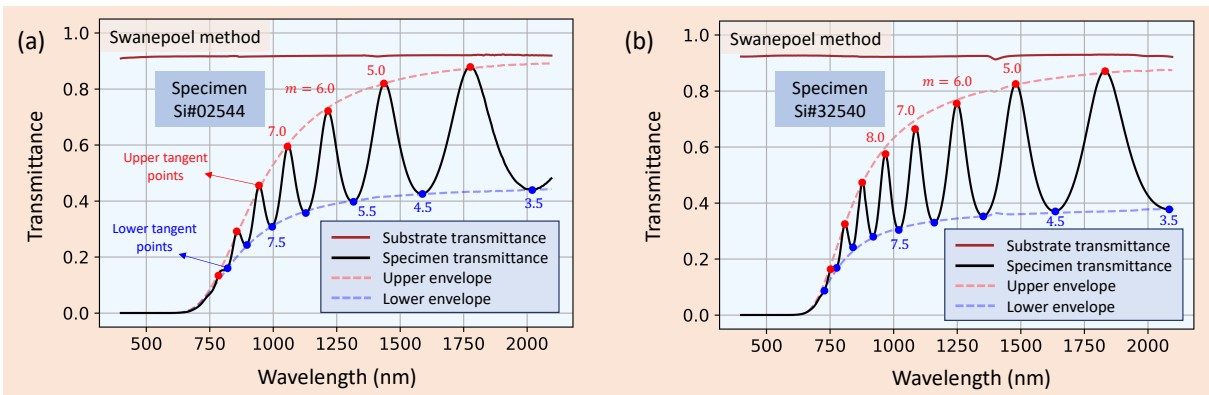

**Figure 6.** Application of the Swanepoel technique as elaborated in the text: Using the as-measured normal-incidence transmission spectra of (**a**) Si#02544 and (**b**) Si#32540 *a*-Si thin-film samples, and that of the finite glass substrate alone. The two constructed, top- and bottom-transmission envelopes, are also displayed for the two *a*-Si specimens. Moreover, some *exact* values of the interference order, $m$, are indicated in this figure in order to illustrate the model-free Swanepoel technique.

Lastly, the absorption coefficient, $\alpha$, belonging to the thin-layer material can now be determined by using the expression $x_{\text{uni}} = e^{-\alpha d_{\text{uni}}}$, where $x_{\text{uni}}$ is the absorbance associated in this case with a constant (uniform) layer thickness, $d_{\text{uni}}$ ($\equiv \bar{d}_{\text{acc}}$), and it is expressed by the following equation:

$$x_{\text{uni}} = \frac{M_+ - \sqrt{M_+^2 - (n^2 - 1)^3 (n^2 - s^4)}}{(n-1)^3 (n - s^2)},$$ (29)

where

$$M_+ = \frac{8n^2 s}{T_+} + \left(n^2 - 1\right)\left(n^2 - s^2\right).$$ (30)

We have used the top-envelope curve, $T_+$, with Equation (29), in order to more accurately obtain the values of $\alpha$ over the spectral regions of weak and medium absorption.

Figure 6 displays the typical room-temperature-measured, unshrunk transmission spectra for two representative, as-deposited *a*-Si samples, as a function of vacuum wave-

length. A key observation from this figure is the existence of a clear Fabry–Perot interference fringe pattern in the transmittance spectra. The breadth of the FP interference fringes of the transmission spectrum for the Si#32540 sample is greater than that of the spectrum for Si#02544 sample. This is a direct result of the higher values of the refractive index for the Si#32540 thin layer. It is obeyed that the greater the amplitude of the interference fringe pattern of the transmission spectrum, the greater the difference of the refractive index between the weakly absorbing semiconductor film material and the transparent glass substrate, $n - s$.

In Table 3, are indicated all the steps followed in order to carry out the whole optical characterizations of the two representative RFMS-*a*-Si samples, deposited with the particular values of the *average* deposition rate of 1.24 nm/s for the sample Si#02544, and 1.17 nm/s for the specimen Si#32540. By starting up from the normal-incidence transmittance spectra again displayed in Figure 6, and by rigorously using the *complete* algorithm proposed by Swanepoel step-by-step, Table 3 was finally created.

**Table 3.** Optical and geometrical characterizations method for the two representative samples, Si#02544 and Si#32540, grown by using the highest Ar working-gas pressures of 4.4 and 4.0 Pa, respectively. Values of the optical and geometrical parameters, $\lambda_{tan}, T_s, s, T_+, T_-, n_{crude}, d_{crude}, m_0, m, d_{acc}, \alpha,$ and $k$, calculated from their respective normal-incidence transmittance spectra, by employing to that end the Swanepoel transmission-envelope approach. The meaning of all the symbols are fully explained in the text. The values of $T_+$ and $T_-$ in bold character refer to the actual 'measured' value of transmission, while the other one belongs to the calculated envelope.

**Si#02544**

| $\lambda_{tan}$ (nm) | $T_s$ | $s$ | $T_+$ | $T_-$ | $n_{crude}$ | $d_{crude}$ (nm) | $m_0$ | $m$ | $d_{acc}$ (nm) | $n_{acc}$ | $x$ | $\alpha$ ($\times 10^3$ cm$^{-1}$) | $k$ |
|---|---|---|---|---|---|---|---|---|---|---|---|---|---|
| 2020 | 0.9210 | 1.509 | 0.8896 | **0.4395** | 3.162 | N.A. | 3.50 | **3.5** | 1118 | **3.161** | 0.976 | 0.213 | **0.003** |
| 1776 | 0.9204 | 1.511 | **0.8731** | 0.4331 | 3.177 | N.A. | 4.01 | **4.0** | 1118 | **3.176** | 0.964 | 0.325 | **0.005** |
| 1588 | 0.9205 | 1.511 | 0.8511 | **0.4244** | 3.194 | 1121 | 4.50 | **4.5** | 1119 | **3.195** | 0.948 | 0.481 | **0.006** |
| 1436 | 0.9177 | 1.523 | **0.8188** | 0.4153 | 3.213 | 1114 | 5.01 | **5.0** | 1117 | **3.210** | 0.924 | 0.703 | **0.008** |
| 1316 | 0.9192 | 1.516 | 0.7819 | **0.4008** | 3.234 | 1120 | 5.50 | **5.5** | 1119 | **3.236** | 0.895 | 0.997 | **0.010** |
| 1216 | 0.9186 | 1.519 | **0.7332** | 0.3844 | 3.257 | 1134 | 6.00 | **6.0** | 1120 | **3.262** | 0.856 | 1.395 | **0.014** |
| 1128 | 0.9181 | 1.521 | 0.6709 | **0.3632** | 3.283 | 1103 | 6.52 | **6.5** | 1117 | **3.278** | 0.803 | 1.958 | **0.018** |
| 1056 | 0.9177 | 1.522 | **0.6003** | 0.3383 | 3.311 | 1094 | 7.02 | **7.0** | 1116 | **3.305** | 0.742 | 2.675 | **0.023** |
| 996 | 0.9170 | 1.526 | 0.5240 | **0.3100** | 3.340 | 1130 | 7.51 | **7.5** | 1118 | **3.339** | 0.671 | 3.566 | **0.029** |
| 944 | 0.9163 | 1.529 | **0.4432** | 0.2777 | 3.370 | 1150 | 7.99 | **8.0** | 1120 | **3.376** | 0.592 | 4.692 | **0.035** |
| 896 | 0.9157 | 1.531 | 0.3572 | **0.2393** | 3.404 | 1121 | 8.51 | **8.5** | 1119 | **3.405** | 0.500 | 6.206 | **0.044** |
| 856 | 0.9171 | 1.525 | **0.2794** | 0.1998 | 3.438 | 1120 | 8.99 | **9.0** | 1120 | **3.444** | 0.410 | 7.980 | **0.054** |
| 820 | 0.9165 | 1.527 | 0.2079 | **0.1595** | 3.474 | 1144 | 9.49 | **9.5** | 1121 | **3.483** | 0.320 | 10.192 | **0.067** |
| 784 | 0.9165 | 1.528 | **0.1400** | 0.1156 | 3.516 | 1068 | 10.04 | **10.0** | 1115 | **3.505** | 0.226 | 13.308 | **0.083** |
| | | | | | $\bar{d}_{crude} = 1120$ nm; | | $\sigma_{crude} = 24$ nm (2.2%); | | | $\bar{d}_{acc} = \mathbf{1118}$ nm; | | $\sigma_{acc} = \mathbf{2}$ nm (**0.2%**) | | |

**Si#32540**

| $\lambda_{tan}$ (nm) | $T_s$ | $s$ | $T_+$ | $T_-$ | $n_{crude}$ | $d_{crude}$ (nm) | $m_0$ | $m$ | $d_{acc}$ (nm) | $n_{acc}$ | $x$ | $\alpha$ ($\times 10^3$ cm$^{-1}$) | $k$ |
|---|---|---|---|---|---|---|---|---|---|---|---|---|---|
| 2084 | 0.9231 | 1.500 | 0.8748 | **0.3799** | 3.464 | N.A. | 3.52 | **3.5** | 1053 | **3.462** | 0.966 | 0.328 | **0.005** |
| 1832 | 0.9297 | 1.472 | **0.8650** | 0.3699 | 3.478 | N.A. | 4.02 | **4.0** | 1053 | **3.479** | 0.955 | 0.436 | **0.006** |
| 1636 | 0.9286 | 1.476 | 0.8458 | **0.3647** | 3.495 | 1055 | 4.52 | **4.5** | 1053 | **3.495** | 0.942 | 0.565 | **0.007** |
| 1480 | 0.9259 | 1.488 | **0.8224** | 0.3597 | 3.513 | 1052 | 5.03 | **5.0** | 1053 | **3.513** | 0.927 | 0.719 | **0.009** |
| 1352 | 0.9250 | 1.492 | 0.7970 | **0.3525** | 3.534 | 1046 | 5.54 | **5.5** | 1052 | **3.530** | 0.909 | 0.906 | **0.010** |
| 1248 | 0.9238 | 1.497 | **0.7679** | 0.3444 | 3.557 | 1050 | 6.04 | **6.0** | 1053 | **3.555** | 0.888 | 1.126 | **0.011** |
| 1160 | 0.9230 | 1.500 | 0.7344 | **0.3347** | 3.582 | 1055 | 6.54 | **6.5** | 1052 | **3.579** | 0.863 | 1.396 | **0.013** |
| 1084 | 0.9222 | 1.504 | **0.6945** | 0.3232 | 3.610 | 1041 | 7.05 | **7.0** | 1051 | **3.602** | 0.833 | 1.740 | **0.015** |
| 1020 | 0.9218 | 1.505 | 0.6488 | **0.3098** | 3.641 | 1039 | 7.56 | **7.5** | 1051 | **3.631** | 0.796 | 2.165 | **0.018** |
| 968 | 0.9217 | 1.506 | **0.5993** | 0.2949 | 3.671 | 1082 | 8.03 | **8.0** | 1055 | **3.676** | 0.756 | 2.661 | **0.021** |
| 920 | 0.9220 | 1.504 | 0.5395 | **0.2762** | 3.706 | 1089 | 8.53 | **8.5** | 1055 | **3.712** | 0.703 | 3.340 | **0.025** |
| 876 | 0.9221 | 1.504 | **0.4686** | 0.2532 | 3.745 | 1036 | 9.05 | **9.0** | 1053 | **3.743** | 0.637 | 4.281 | **0.030** |
| 840 | 0.9239 | 1.497 | 0.3969 | **0.2274** | 3.784 | 1049 | 9.54 | **9.5** | 1054 | **3.788** | 0.566 | 5.412 | **0.036** |
| 808 | 0.9241 | 1.496 | **0.3221** | 0.1986 | 3.826 | 1089 | 10.03 | **10.0** | 1056 | **3.836** | 0.484 | 6.881 | **0.044** |
| 776 | 0.9245 | 1.494 | 0.2395 | **0.1620** | 3.875 | 1024 | 10.57 | **10.5** | 1051 | **3.868** | 0.384 | 9.0929 | **0.056** |
| 752 | 0.9251 | 1.491 | **0.1761** | 0.1292 | 3.919 | 1050 | 11.03 | **11.0** | 1056 | **3.927** | 0.299 | 11.452 | **0.069** |
| 728 | 0.9260 | 1.487 | 0.1161 | **0.0931** | 3.968 | 1093 | 11.54 | **11.5** | 1055 | **3.974** | 0.210 | 14.824 | **0.086** |
| | | | | | $\bar{d}_{crude} = 1059$ nm; | | $\sigma_{crude} = 23$ nm; (2.2%) | | | $\bar{d}_{acc} = \mathbf{1053}$ nm; | | $\sigma_{acc} = \mathbf{2}$ nm (**0.2%**) | | |

N.A.: Not applicable.

## 8. Concluding Remarks

We have devised a new procedure to generate an analytic complex dielectric function from a non-analytic dispersion model. We can write the transformation as an integral form that aims to smooth the chosen optical function by convolving with a Lorentzian-type function. This function has a half-width broadening parameter $\Gamma$ that we can fit to be as small as necessary without ever reaching a zero energy value.

The transforming procedure has been applied to the original well-known NTLU dispersion model described by a piecewise mathematical function. This particular approach involves a change in the functionality around the specific value of the energy gap, $E_g$. We have also carried out a relevant modification in the expression for the Urbach-band-tail part of the TLU model, in order to avoid the existing divergence at zero-photon energy of the NTLU model. The 'analyticized' TLU model was adopted in order to accurately calculate the optical properties of RFMS-*a*-Si thin layers, grown at both room temperature and 325 °C, by only employing the normal-incidence UV/visible/NIR transmission spectra of the specimens, and the inverse-synthesis approach in order to extract the optical constants.

Last but not least, it ought to be noted that the fitted values of the bandgap parameter corresponding to the NTLU function are either smaller or larger than those for the novel ATLU function, depending upon the deposition temperature employed as displayed in Table 1. In addition, the calculated values of the Urbach energy parameter corresponding to the NTLU function are not significantly influenced by the deposition temperature in all the cases studied. On the contrary, it has been found by choosing the holomorphic dielectric function that the parameter $E_u$ notably decreases as the deposition temperature increases.

The lack of analyticity in the original NTLU function, and its associated intrinsic inaccuracy (in contrast with the holomorphic ATLU function), gives rise to the lack of agreement between the best-fit values of the two analyzed TLU parameters, $E_g$ and $E_u$, respectively, belonging to the two different optical functions. In conclusion, it is worth noting that all the presented findings represent a step forward in improving the quality of the optical-and-geometrical characterizations of the amorphous semiconductor thin films, by exclusively using their normal-incidence spectral transmittance.

**Author Contributions:** M.B., E.M., Methodology: M.B., A.P.M., E.M.; Software: M.B., A.P.M.; Visualization: M.B., M.G., E.B.; Writing—review & edting: M.B., M.G., A.P.M., F.W., E.M. Validation: M.G., A.P.M., D.M., A.K.K., O.C., F.W.; Data curation: E.B., S.M.F.; Investigation: E.B., S.M.F. Supervision: D.M., A.K.K., O.C., F.W.; Formal analysis: E.M.; Project administration: E.M. All authors have read and agreed to the published version of the manuscript.

**Funding:** This research received no external funding.

**Conflicts of Interest:** The authors declare no conflict of interest.

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
