# Peer review of "Application of the Holomorphic Tauc-Lorentz-Urbach Function to Extract the Optical Constants of Amorphous Semiconductor Thin Films"

_coatings, doi:10.3390/coatings12101549_

Round 1

Reviewer 1 Report

This manuscript is presenting an interesting topic on develoing a dielectric function using Tauc-Lorentz-Urbach (TLU) dispersion model.

the few comments and questions are due before we accept this manuscript:

1)how do you satisfy the continuity of the new weight function and its first derivative in eq. 7.

2)is the eq. 20 valid for thighly non-uniform thin layers? 

3)in introduction, citation is requried to https://doi.org/10.3390/coatings11080945

4) in fig. 6: what's the impact of greater n difference between weakly-absorbing semiconductor film material and transparent glass substrate?

when authors take my comments in and reply to my questions, i can reconsider my decision. no comment must be skipped or the paper is rejected.

Reviewer 2 Report

In this paper, an analytic dispersion model, based on the Tauc-Lorentz-Urbach (TLU) dispersion model, is developed and applied to the optical analysis of sputtered amorphous silicon thin films deposited at different temperatures. The optical analysis of the amorphous semiconductor thin films is made from their normal-incidence transmission spectra. The results were compared with those obtained from the non-analytic dispersion model. The results show that the analytic model provides fitting parameters more consistent than those from the non-analytic model, particularly the Urbach Energy.

The paper is very interesting because of the consistency of the information provided by the analytical model.

Authors just should include more discussion about the values of the FoM reported in Table 1. It is important to know what is the limit value of this FoM to consider that the fit to experimental data is good.
